# Investigation of anxiety levels of 1637 healthcare workers during the epidemic of COVID-19

**Meiping Shen[1], Hongzhen Xu[1], Junfen Fu👤[1]\*, Tianlin Wang[1], Zangzang Fu[1], Xiaomei Zhao[2], Gendi Zhou[3], Qi Jin[4], Guixiu Tong[5]**

**1** The Children's Hospital, Zhejiang University School of Medicine, National Clinical Research Center for Child Health, Hangzhou, China, **2** The First Affiliated Hospital, Zhejiang University School of Medicine, Hangzhou, China, **3** Hangzhou Hospital of Traditional Chinese Medicine, Hangzhou, China, **4** Xiaoshan First People's Hospital, Hangzhou, China, **5** Chunan Hospital of Traditional Chinese Medicine, Zhejiang University School of Medicine, Hangzhou, China

\* fjf68@zju.edu.cn

**Data Availability Statement:** All relevant data are within the manuscript and its Supporting information files.

**Funding:** This work was supported by Zhejiang University special research scientific research fund

## Abstract

### Objective

This study aimed to investigate the anxiety levels of healthcare workers and to provide guidance on potential accurate social and psychological interventions for healthcare workers during the epidemic of COVID-19 in Zhejiang Province, China.

### Methods

Healthcare workers from five hospitals in Zhejiang Province were randomly selected into this study. Zung Self-Assessment Scale for Anxiety (SAS) was used to evaluate the anxiety status of the included 1637 healthcare workers.

### Results

The total anxiety score of healthcare workers in Zhejiang Province was 30.85 ± 6.89. The univariate analysis showed that the anxiety level of healthcare workers was related to gender, education, occupation, physical condition, job risk coefficient, and with family members on the first-line combating COVID-19 (P <0.05). The multivariate analysis showed that physical condition and job risk coefficient were predictors of anxiety levels of healthcare workers.

### Conclusions

During the epidemic of COVID-19, 1637 healthcare workers generally had an increased tendency to have anxiety. Individualized assessment of the anxiety level of healthcare workers should be provided, and different interventions should be given based on the evaluation results.

for COVID-19 prevention and control. The funders had no role in study design, data collection and analysis, decision to publish, or preparation of the manuscript.

**Competing interests:** The authors have declared that no competing interests exist.

**Abbreviations:** SAS, Self-Assessment Scale for Anxiety.

# 1 Introduction

When people are confronted with events that are dangerous and life threatening, or situations that need great efforts, they will feel nervous and unpleasant. This emotion is anxiety. Moderate anxiety can increase work motivation, while excessive anxiety is a kind of pathological emotion. The pathological emotion directly affects people's mental health [1]. The outbreak of COVID-19 at the beginning of 2020 is a public health disaster. The epidemic of COVID-19 has a huge psychological impact on general population [2], workers [3], psychiatric patients [4] and rural population [5]. Under this background, healthcare workers, especially the first line healthcare workers, are faced with unimaginable challenges [6]. They need to overcome the fear of infection at any time during the work. They need to focus on treating and comforting patients even though they are suffering from physical and mental exhaustion. And many frontline healthcare workers have been or are being infected during this pandemic. Meanwhile, the anxiety status in healthcare workers will lead to the decline of their immunity. Therefore, it is important for providing psychological counseling for healthcare workers. The purpose of this study was to investigate the anxiety levels of 1 637 healthcare workers during the epidemic of COVID-19 in Zhejiang Province, China.

# 2 Methods

## 2.1 Design and subjects

During the epidemic of COVID-19, the subjects were healthcare workers in Zhejiang Province, including two provincial hospitals, one municipal hospital, one municipal hospital of traditional Chinese medicine and one county hospital. A total of 1 648 questionnaires were sent out, 1637 of which were valid questionnaires. The institutional review board of the Zhejiang University Children's Hospital approved the study and waived the requirement for obtaining informed consent due to the retrospective nature of this study (2020-IRB-059). The data were analyzed anonymously.

## 2.2 Survey tools

General information included gender, education, occupation, age, family income, physical condition, contact history, post risk coefficient, whether immediate family infected, whether immediate family on the first-line to COVID-19. The questionnaire was distributed randomly to healthcare workers in the five hospitals. The anxiety levels of 1 637 healthcare workers was assessed by Zung Self-Assessment Scale (SAS), which was widely used to evaluate the anxiety status of subjects. There were 20 items in the scale, each were scored by 4 grades. The items 5, 9, 13, 17 and 19 were reverse scores. The original score was multiplied by 1.25 and the integral part was the standard score. The mild anxiety score was 50–59 points, the moderate anxiety score was 60–69 points and the severe anxiety score was ≥70 points. The split half reliability of the scale was 0.696, the retest reliability was 0.777, and the internal consistency reliability was 0.826.

## 2.3 Data collection

The project was approved by the ethics committee of the hospitals and was given the consent by the nursing department of each hospital. The purpose, significance and filling method of the questionnaire were explained to the head nurses under unified instruction by the project researchers. As research coordinators, the trained head nurses sent out the electronic questionnaire to the nursing staff and guided them to fill in the questionnaire. Data collection took

place over three days and the validity of the questionnaire was checked by the project research-ers. The questionnaires with missing items or regular filling were eliminated.

## 2.4 Statistical analyses

All data were analyzed with SPSS26.0 statistical software. Metrological data were expressed as mean ± standard deviation. Independent sample analysis was conducted by t-test, and multi-group comparison was conducted by one-way analysis of variance and multivariate linear regression analysis. A value of $P < 0.05$ was considered as statistical significant.

## 3 Results

### 3.1 Total anxiety score of healthcare workers during the epidemic of COVID-19

The total anxiety score of healthcare workers in Zhejiang Province was 30.85 ± 6.89. The scale of anxiety level is shown in Table 1.

### 3.2 Univariate analysis of 1637 healthcare workers' anxiety level

Statistically significant differences were found in gender, education, occupation, physical con-dition, contact history, post risk coefficient, whether with family members on the first-line combating COVID-19 (P<0.05). The anxiety level of females was higher than that of males; and nurses had higher score than doctors. Healthcare workers with undergraduate degree was higher than those with secondary technical school degree, or master degree, or doctor degree. Those with underlying diseases, common cold or flu were higher than healthy staff. Those with family members on the first-line were higher than those without. There was a positive correlation between job risk coefficient and anxiety level. (Table 2).

### 3.3 Multivariate analysis of 1637 healthcare workers' anxiety level

Predictors captured in univariate analysis were enrolled in multivariate analysis. Set up the sub-variables for the multivariate category disordered independent variables (Table 3). Multi-ple linear regression analysis was conducted according to the introduction model of $\alpha = 0.05$. The results showed that physical condition and post risk coefficient were predictors of anxiety level (P < 0.05). (Table 4).

## 4 Discussion

As a new virus, COVID-19 was highly infectious and susceptible to human beings. In addition to pneumonia, it may develop multiple organ dysfunction, with a mortality rate of 2.30%. Cur-rently, there was still no specific treatment or preventive vaccine. During the epidemic of COVID-19, the frontline healthcare workers against COVID-19 built a safe fort, and relieved the psychological pressure of patients. However, healthcare workers themselves were very likely to have anxiety and posttraumatic stress syndrome. Literature revealed that SARS in

**Table 1. Anxiety status of 1637 healthcare workers.**

| Anxiety level | N(%) | Score |
|---|---|---|
| None | 1473(90.0) | 36.54 ± 6.12 |
| Mild | 126(7.7) | 53.51 ± 2.63 |
| Moderate | 31(1.9) | 63.34 ± 2.42 |
| Severe | 7(0.4) | 79.10 ± 6.32 |

**Table 2. Univariate analysis of healthcare workers' anxiety level (n = 1637).**

| Variables | | N | Score | T /F | P |
|---|---|---|---|---|---|
| Gender | Male | 166 | 29.49±5.712 | 7.073[1] | 0.008 |
| | Female | 1471 | 30.98±6.99 | | |
| Age | 20~30 | 593 | 30.32±6.583 | 2.137[2] | 0.059 |
| | 31~40 | 739 | 31.20±7.18 | | |
| | 41~45 | 237 | 31.22±6.72 | | |
| | 46~50 | 64 | 30.34±6.47 | | |
| | >50 | 4 | 25.75±5.62 | | |
| Education level | Technical secondary school | 23 | 28.65±4.46 | 5.933[2] | 0.000 |
| | Junior | 178 | 30.97±7.56 | | |
| | Undergraduate | 1285 | 31.12±6.93 | | |
| | Master | 125 | 28.86±5.37 | | |
| | Doctor | 26 | 26.88±5.30 | | |
| Occupation | Doctor | 212 | 28.74±5.72 | 8.410[2] | 0.000 |
| | Nurse | 1383 | 31.19±7.00 | | |
| | Technician | 11 | 29.00±4.79 | | |
| | Others | 29 | 29.72±7.12 | | |
| Family monthly income (RMB) | <8000 | 475 | 31.25±7.49 | 1.970[2] | 0.117 |
| | 10000~20000 | 817 | 30.90±6.76 | | |
| | 20000~30000 | 290 | 30.18±6.37 | | |
| | >50000 | 55 | 29.71±5.42 | | |
| Physical condition | Health | 1562 | 30.70±6.89 | 6.372[2] | 0.002 |
| | Common cold or influenza | 12 | 32.75±6.41 | | |
| | Other | 63 | 33.73±6.04 | | |
| Post risk coefficient | High | 448 | 32.49±7.76 | 20.615[2] | 0.000 |
| | Medium | 646 | 30.96±6.58 | | |
| | Low | 437 | 29.68±6.04 | | |
| | Extremely low | 106 | 27.75±6.09 | | |
| Contact history | Yes | 59 | 31.73±8.47 | 1.037[1] | 0.309 |
| | No | 1578 | 30.80±6.82 | | |
| Family members on the first-line combating COVID-19 | Yes | 120 | 32.19±7.83 | 5.059[1] | 0.025 |
| | No | 1517 | 30.73±6.79 | | |

[1] T value;

[2] F value.

**Table 3. Assignment of independent variables.**

| Variables | Score points |
|---|---|
| Gender | 1 = male; 2 = female |
| Education | 1 = technical secondary school; 2 = junior; 3 = undergraduate; 4 = master; 5 = doctor |
| Occupation | 1 = doctor; 2 = nurse; 3 = technician; 4 = others |
| Physical condition | 1 = healthy; 2 = common cold or flu; 3 = others |
| Post risk coefficient | 1 = extremely low; 2 = low; 3 = medium; 4 = high |
| Family members on the first-line combating COVID-19 | 1 = yes; 2 = no |

**Table 4. Multiple linear regression analysis of healthcare workers' anxiety level (n = 1637).**

| Variables | Regression coefficient | Standard error | Standard regression coefficient | t | P |
|---|---|---|---|---|---|
| Constant | 24.302 | 2.302 | - | 10.556 | 0.000 |
| Physical condition | 1.432 | 0.424 | 0.082 | 3.381 | 0.001 |
| Post risk coefficient | 1.441 | 0.188 | 0.185 | 7.654 | 0.000 |

2003, Wenchuan earthquake in 2008, Ebola virus in 2014, and MERS, all these emergency events caused different degrees of psychological trauma to the healthcare workers, including anxiety, insomnia, depression and emotional disorders [7–10]. With more infected cases as well as death cases in the COVID-19 epidemic, healthcare workers have been facing more challenges physically and psychologically [11]. The Chinese government attached great importance to the mental health of medical personnel. At the national level, many measures had been taken to relieve the pressure of medical personnel. In the early stage, the psychological counseling hotline was opened. The psychological doctors, social workers, volunteers and government staff were organized to help the front-line personnel, so as to reduce the negative effect caused by the outbreak to medical personnel to minimum.

This study showed that the score of anxiety level of healthcare workers in Zhejiang Province was high, indicating that healthcare workers in Zhejiang Province had a tendency of anxiety due to the impact of COVID-19 outbreak. Participating in public emergency events was the most common psychological stress response for healthcare workers, which was a complex stressor anxiety with biological, psychological, social and other properties [12]. This may be related to the severity and unknown tendency of the outbreak of COVID-19. No infections among healthcare workers in Zhejiang have been reported. However, along with returning to work and huge population flow, the asymptomatic infected people may have impact on the anxiety level of healthcare workers.

The results showed that there were significant differences in gender, education level, occupation, physical condition, post risk coefficient, and, whether with family members on the first-line to COVID-19 (P<0.05). Females had higher score than males, which may be related to females' higher anxiety tendency than male [13]. Nurses had higher score than doctors, with one possible explanation was that nurses needed to contact patients for longer time to complete medical tasks [3]. Nurses were more likely to contact patients at the first time and undertake a large number of nursing tasks. The probability of infection was higher than doctors. Their emotions were easy to fluctuate. The anxiety level in those with undergraduate degree was relatively high, which may be related to the cognition of COVID-19. The healthcare workers with underlying diseases were more likely to have anxiety than those without underlying diseases, which may be related to concern of their low immunity. Additionally, the post risk coefficient was positively related to the anxiety tendency of the healthcare workers. The reason may be that the healthcare workers in these posts needed to bear greater psychological pressure, higher intensity medical care tasks, heavier protective measures, higher risk of infection, long-term separation from family members, lack of protective materials, etc. The anxiety tendency of those healthcare workers with family members on the first-line to COVID-19 was higher than those without. This may be explained by that family relationship directly affected the anxiety level of family members.

During this investigation, the epidemic and tendency were still not clear, there were many unknown areas, and there were regional limitations in sample collection. Further follow-up was needed. Individualized evaluation of anxiety levels of healthcare workers and individualized interventions measures should be taken. Mild anxiety, mainly psychological intervention, such as relaxation training, cognitive behavior therapy, physical exercise and fitness, moderate

reduction of clinical work load, arrangement of rotation, etc. Moderate anxiety, in addition to the measures of psychological intervention and social support, can consider the treatment of anti-anxiety drugs. Severe anxiety, first consider the clinical anti-anxiety drugs, combining psychological intervention and social support.

The investigation of anxiety level of healthcare workers in Zhejiang Province may provide a scientific basis for the formulation of effective psychosocial intervention strategies. Accurate intervention dependent on the individualized evaluation of anxiety levels should be taken. Psychological intervention, such as relaxation training, cognitive behavior therapy, physical exercise and fitness, moderate reduction of clinical work load, arrangement of rotation, etc., may be the preferred therapy for mild anxiety. While the additional social support is necessary for the moderate anxiety, and the clinical anti-anxiety drugs are firstly considered for severe anxiety. In addition, other factors included gender, education, occupation, age, family, income, physical condition, contact history, post risk coefficient, etc., should be taken into consideration.

Though the results demonstrated the anxiety levels from large sample size of healthcare workers, this study has several limitations. First, it is difficult to make causal interpretation from the cross-sectional investigation. Second, there were regional limitations in sample collection. Third, this study was focused on the assessment of anxiety levels, other psychological disorders such as depression or acute stress were not included in the survey. While the prevalence of depression or acute stress may impact anxiety levels and affect the results interpretation.

In this study, we demonstrated the total anxiety score of 1637 healthcare workers in Zhejiang Province, China. The anxiety level was related to gender, education, occupation, physical condition, job risk coefficient, and with family members on the first-line combating COVID-19. In addition, the physical condition and job risk coefficient were predictors of anxiety levels of healthcare workers. We recommend that accurate psychological interventions should be given based on the individualized assessment of the anxiety level of healthcare workers.

## Supporting information

**S1 File. Data from Chunan Hospital of Traditional Chinese Medicine, Zhejiang University School of Medicine.**
(XLS)

**S2 File. Data from Hangzhou Hospital of Traditional Chinese Medicine.**
(XLS)

**S3 File. Data from the Children's Hospital, Zhejiang University School of Medicine, National Clinical Research Center for Child Health.**
(XLS)

**S4 File. Data from the First Affiliated Hospital, Zhejiang University School of Medicine.**
(XLS)

**S5 File. Data from Xiaoshan First People's Hospital.**
(XLS)

## Acknowledgments

Thanks to professor Meili Lou in psychology teaching and research group, medical college, Hangzhou Normal University, Zhejiang Province, China.

## Author Contributions

**Conceptualization:** Meiping Shen, Junfen Fu.

**Data curation:** Meiping Shen, Hongzhen Xu, Tianlin Wang, Zangzang Fu, Xiaomei Zhao, Gendi Zhou, Qi Jin, Guixiu Tong.

**Formal analysis:** Meiping Shen, Zangzang Fu, Xiaomei Zhao, Gendi Zhou, Qi Jin, Guixiu Tong.

**Funding acquisition:** Meiping Shen.

**Investigation:** Meiping Shen, Hongzhen Xu, Tianlin Wang, Zangzang Fu, Xiaomei Zhao, Gendi Zhou, Qi Jin.

**Methodology:** Meiping Shen, Hongzhen Xu, Tianlin Wang, Zangzang Fu, Xiaomei Zhao, Gendi Zhou, Qi Jin, Guixiu Tong.

**Resources:** Meiping Shen.

**Software:** Meiping Shen.

**Supervision:** Junfen Fu.

**Validation:** Meiping Shen, Junfen Fu, Xiaomei Zhao, Qi Jin.

**Writing – original draft:** Meiping Shen, Zangzang Fu.

**Writing – review & editing:** Hongzhen Xu, Junfen Fu, Tianlin Wang.

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
