## [Decision Letter · Decision Letter 0]

2 Sep 2020

PONE-D-20-20196

Investigation of anxiety levels of 1637 healthcare workers during the epidemic of COVID-19

PLOS ONE

Dear Dr. junfen fu,

Thank you for submitting your manuscript to PLOS ONE. After careful consideration, we feel that it has merit but does not fully meet PLOS ONE’s publication criteria as it currently stands. Therefore, we invite you to submit a revised version of the manuscript that addresses the points raised during the review process.

ACADEMIC EDITOR:

We look forward to receiving your revised manuscript.

Kind regards,

Shawky M. Aboelhadid, PhD

Academic Editor

PLOS ONE

Journal Requirements:

2.We suggest you thoroughly copyedit your manuscript for language usage, spelling, and grammar. If you do not know anyone who can help you do this, you may wish to consider employing a professional scientific editing service.  

3.Thank you for stating the following in the Funding Section of your manuscript:

[This work was supported by Zhejiang University special research scientific research fund for COVID-19 prevention and control.]

 [The funders had no role in study design, data collection and analysis, decision to publish, or preparation of the manuscript.]

Reviewers' comments:

Reviewer's Responses to Questions

**Comments to the Author**

1. Is the manuscript technically sound, and do the data support the conclusions?

Reviewer #1: No

Reviewer #2: Yes

2. Has the statistical analysis been performed appropriately and rigorously? 

Reviewer #1: No

Reviewer #2: Yes

3. Have the authors made all data underlying the findings in their manuscript fully available?

Reviewer #1: Yes

Reviewer #2: Yes

4. Is the manuscript presented in an intelligible fashion and written in standard English?

Reviewer #1: No

Reviewer #2: Yes

5. Review Comments to the Author

Reviewer #1: This is an interesting article exploring anxiety levels and mental health interventions and support for healthcare workers from 5 hospitals in China.

Strengths include: large sample size, random selection of participants

There are several comments that need to be addressed:

-Is there any data on prevalence of depression or acute stress as this impacts anxiety levels.

-Line 74/75: Can the authors explain whether the trained head nurse is a research coordinator or whether he/she has a clinical role? Also can authors provide further details about who checked the questionnaire for validity?

“The researcher explained the purpose and the filling method of the questionnaire to the head nurses. The trained head nurses distributed the electronic questionnaire to the nurses and guided them to fill the questionnaire.”

-Line 87, states: “The total anxiety score of healthcare workers in Zhejiang Province was 30.85 ± 6.89, which was higher than that of Chinese norm 9.78 ± 0.46 [4] 87 (t = 6.273, P < 0.01).”

Methodologically why are the authors comparing prevalence of anxiety of “Chinese norm” from an older reference in the Results section and running any comparison test? I presume it is reference 7 rather than 4 as this is mentioned in the Discussion Line 127/128: “This study showed that the score of anxiety level of healthcare workers in Zhejiang Province was higher than that of Chinese norm [7]”. This reference is discussing anxiety levels during the SARS outbreak in Beijing and uses other measures of psychological distress not assessed in the current study. Further please update the reference in Line 87 as it is incorrect: [4] Lehmann M, Bruenahl CA, Addo MM, Becker S, Schmiedel S, Lohse AW, et al. Acute 179 Ebola virus disease patient treatment and health-related quality of life in health care 180 professionals: A controlled study. J Psychosom Res. 2016; 83:69-74 is a study out of Germany and discusses Ebola so it’s unclear why this is being referenced here at all.

-Line 143/144 “Most nurses were women, facing the physiological period under protective clothing. Their emotions were easy to fluctuate.”

This is a sexist statement and is unacceptable.

-Was an intervention done? The abstract introduction is misleading and if no intervention was done this should be revised

“This study aimed to investigate the anxiety levels of healthcare workers and to provide effective and accurate social and psychological interventions for healthcare workers during the epidemic of COVID-19 in Zhejiang Province, China.”

-The discussion section abruptly ends and needs to be revised for cohesion.

Reviewer #2: I have the following comments for the authors to amend. I will review the reply again.

1) The authors stated "The epidemic of COVID-19 has a huge psychological impact on all social groups." The authors need to be more specific and amend as follows:

The epidemic of COVID-19 has a huge psychological impact on general population (Wang et al 2020), workers (Tan et al 2020), psychiatric patients (Hao et al 2020) and rural population (Tran et al 2020).

References:

Wang C, Pan R, Wan X, et al. (2020) A Longitudinal Study on the Mental Health of General Population during the COVID-19 Epidemic in China [published online ahead of print, 2020 Apr 13]. Brain Behav Immun. 2020; S0889-1591(20)30511-0. doi:10.1016/j.bbi.2020.04.028

Tan W, Hao F, McIntyre RS, et al. Is Returning to Work during the COVID-19 Pandemic Stressful? A Study on Immediate Mental Health Status and Psychoneuroimmunity Prevention Measures of Chinese Workforce [published online ahead of print, 2020 Apr 23]. Brain Behav Immun. 2020;S0889-1591(20)30603-6. doi:10.1016/j.bbi.2020.04.055

Hao F, Tan W, Jiang L, et al. Do psychiatric patients experience more psychiatric symptoms during COVID-19 pandemic and lockdown? A Case-Control Study with Service and Research Implications for Immunopsychiatry [published online ahead of print, 2020 Apr 27]. Brain Behav Immun. 2020;S0889-1591(20)30626-7. doi:10.1016/j.bbi.2020.04.069

Tran BX, Phan HT, Nguyen TPT, et al. Reaching further by Village Health Collaborators: The informal health taskforce of Vietnam for COVID-19 responses. J Glob Health. 2020;10(1):010354. doi:10.7189/jogh.10.010354

2) For the following statement, the authors need to cite references:

Under this background, healthcare workers, especially the first line healthcare workers, are faced with unimaginable challenges.

Reference:

Tan BYQ, Chew NWS, Lee GKH, et al. Psychological Impact of the COVID-19 Pandemic on Health Care Workers in Singapore [published online ahead of print, 2020 Apr 6]. Ann Intern Med. 2020;M20-1083. doi:10.7326/M20-1083

3) Under the discussion, the authors stated "Literature revealed that SARS in

117 2003, Wenchuan earthquake in 2008, Ebola virus in 2014, and MERS, all these emergency

118 events caused different degrees of psychological trauma to the healthcare workers, including

anxiety, insomnia, depression and emotional disorders [2-6]"

Please compare their findings with COVID-19 study instead:

Chew NWS, Lee GKH, Tan BYQ, et al. A multinational, multicentre study on the psychological outcomes and associated physical symptoms amongst healthcare workers during COVID-19 outbreak [published online ahead of print, 2020 Apr 21]. Brain Behav Immun. 2020;S0889-1591(20)30523-7. doi:10.1016/j.bbi.2020.04.049

6. PLOS authors have the option to publish the peer review history of their article (what does this mean?). If published, this will include your full peer review and any attached files.

Reviewer #1: No

Reviewer #2: **Yes: **Roger Ho

---

## [Author Response · Author response to Decision Letter 0]

30 Oct 2020

Response to reviewers’ comments

Reviewer #1: This is an interesting article exploring anxiety levels and mental health interventions and support for healthcare workers from 5 hospitals in China.

Strengths include: large sample size, random selection of participants

There are several comments that need to be addressed:

-Is there any data on prevalence of depression or acute stress as this impacts anxiety levels.

Reply: 

Thanks for reviewer's comments. This study was focused on the assessment of anxiety levels of 1637 healthcare workers, and the data on prevalence of depression or acute stress were not included in the study. We agree that prevalence of depression or acute stress may impact anxiety levels. We have discussed it as one of the limitations of this study in the discussion section as below: 

- Page 13, line 177-179: "Third, this study was focused on the assessment of anxiety levels, other psychological disorders such as depression or acute stress were not included in the survey. While the prevalence of depression or acute stress may impact anxiety levels and affect the results interpretation".

-Line 74/75: Can the authors explain whether the trained head nurse is a research coordinator or whether he/she has a clinical role? Also can authors provide further details about who checked the questionnaire for validity?

“The researcher explained the purpose and the filling method of the questionnaire to the head nurses. The trained head nurses distributed the electronic questionnaire to the nurses and guided them to fill the questionnaire.”

Reply: 

- In this project, the trained head nurse serves as a research coordinator.

- Project researchers, who conducted and analyze the survey, checked the questionnaire for validity.

 We have revised the descriptions in the manuscript as below:

- Page 4-5, "Data collection", line 75-81, “The project was approved by the ethics committee of the hospitals and was given the consent by the nursing department of each hospital. The purpose, significance and filling method of the questionnaire were explained to the head nurses under unified instruction by the project researchers. As research coordinators, the trained head nurses sent out the electronic questionnaire to the nursing staff and guided them to fill in the questionnaire. Data collection took place over three days and the validity of the questionnaire was checked by the project researchers. The questionnaires with missing items or regular filling were eliminated.”

-Line 87, states: “The total anxiety score of healthcare workers in Zhejiang Province was 30.85 ± 6.89, which was higher than that of Chinese norm 9.78 ± 0.46 [4] 87 (t = 6.273, P < 0.01).”

Methodologically why are the authors comparing prevalence of anxiety of “Chinese norm” from an older reference in the Results section and running any comparison test? I presume it is reference 7 rather than 4 as this is mentioned in the Discussion Line 127/128: “This study showed that the score of anxiety level of healthcare workers in Zhejiang Province was higher than that of Chinese norm [7]”. This reference is discussing anxiety levels during the SARS outbreak in Beijing and uses other measures of psychological distress not assessed in the current study. Further please update the reference in Line 87 as it is incorrect: [4] Lehmann M, Bruenahl CA, Addo MM, Becker S, Schmiedel S, Lohse AW, et al. Acute 179 Ebola virus disease patient treatment and health-related quality of life in health care 180 professionals: A controlled study. J Psychosom Res. 2016; 83:69-74 is a study out of Germany and discusses Ebola so it’s unclear why this is being referenced here at all.

Reply: 

We thank reviewer for pointing this out. The manuscript has been revised as below: 

- Comparison prevalence of anxiety of “Chinese norm” from an older reference in the Results section has been deleted from the result and discussion section.

- Comparison test (t test) has been deleted from the result. 

- Incorrect reference [4] has been deleted.

-Line 143/144 “Most nurses were women, facing the physiological period under protective clothing. Their emotions were easy to fluctuate.”

This is a sexist statement and is unacceptable.

Reply: 

- The sentence has been deleted in the revision.

-Was an intervention done? The abstract introduction is misleading and if no intervention was done this should be revised

Reply: 

We have revised the descriptions in the abstract section as below: 

- Page 2, line 19-21: "This study aimed to investigate the anxiety levels of healthcare workers and to provide guidance on potential accurate social and psychological interventions for healthcare workers during the epidemic of COVID-19 in Zhejiang Province, China ".

-The discussion section abruptly ends and needs to be revised for cohesion.

Reply: 

We have revised the end of the discussion section and added the conclusion section in the revision as below: 

- Page 12-13, line 165-179: "The investigation of anxiety level of healthcare workers in Zhejiang Province may provide a scientific basis for the formulation of effective psychosocial intervention strategies. Accurate intervention dependent on the individualized evaluation of anxiety levels should be taken. Psychological intervention, such as relaxation training, cognitive behavior therapy, physical exercise and fitness, moderate reduction of clinical work load, arrangement of rotation, etc., may be the preferred therapy for mild anxiety. While the additional social support is necessary for the moderate anxiety, and the clinical anti-anxiety drugs are firstly considered for severe anxiety. In addition, other factors included gender, education, occupation, age, family, income, physical condition, contact history, post risk coefficient, etc., should be taken into consideration. 

 Though the results demonstrated the anxiety levels from large sample size of healthcare workers, this study has several limitations. First, it is difficult to make causal interpretation from the cross-sectional investigation. Second, there were regional limitations in sample collection. Third, this study was focused on the assessment of anxiety levels, other psychological disorders such as depression or acute stress were not included in the survey. While the prevalence of depression or acute stress may impact anxiety levels and affect the results interpretation.".

- Page 13, Conclusion, line 180-185: "In this study, we demonstrated the total anxiety score of 1637 healthcare workers in Zhejiang Province, China. The anxiety level was related to gender, education, occupation, physical condition, job risk coefficient, and with family members on the first-line combating COVID-19. In addition, the physical condition and job risk coefficient were predictors of anxiety levels of healthcare workers. We recommend that accurate psychological interventions should be given based on the individualized assessment of the anxiety level of healthcare workers. ".

Reviewer #2: I have the following comments for the authors to amend. I will review the reply again.

1) The authors stated "The epidemic of COVID-19 has a huge psychological impact on all social groups." The authors need to be more specific and amend as follows:

The epidemic of COVID-19 has a huge psychological impact on general population (Wang et al 2020), workers (Tan et al 2020), psychiatric patients (Hao et al 2020) and rural population (Tran et al 2020).

References:

Wang C, Pan R, Wan X, et al. (2020) A Longitudinal Study on the Mental Health of General Population during the COVID-19 Epidemic in China [published online ahead of print, 2020 Apr 13]. Brain Behav Immun. 2020; S0889-1591(20)30511-0. doi:10.1016/j.bbi.2020.04.028

Tan W, Hao F, McIntyre RS, et al. Is Returning to Work during the COVID-19 Pandemic Stressful? A Study on Immediate Mental Health Status and Psychoneuroimmunity Prevention Measures of Chinese Workforce [published online ahead of print, 2020 Apr 23]. Brain Behav Immun. 2020;S0889-1591(20)30603-6. doi:10.1016/j.bbi.2020.04.055

Hao F, Tan W, Jiang L, et al. Do psychiatric patients experience more psychiatric symptoms during COVID-19 pandemic and lockdown? A Case-Control Study with Service and Research Implications for Immunopsychiatry [published online ahead of print, 2020 Apr 27]. Brain Behav Immun. 2020;S0889-1591(20)30626-7. doi:10.1016/j.bbi.2020.04.069

Tran BX, Phan HT, Nguyen TPT, et al. Reaching further by Village Health Collaborators: The informal health taskforce of Vietnam for COVID-19 responses. J Glob Health. 2020;10(1):010354. doi:10.7189/jogh.10.010354

Reply: 

Thanks for reviewer's comments. We have revised the descriptions and included the references in the revision as below: 

- Page 3, line 42-44: "The epidemic of COVID-19 has a huge psychological impact on general population [2], workers [3], psychiatric patients [4] and rural population [5]".

References: 

[2] Wang C, Pan R, Wan X, et al. A Longitudinal Study on the Mental Health of General Population during the COVID-19 Epidemic in China. Brain Behav Immun. 2020; S0889-1591(20)30511-0. doi:10.1016/j.bbi.2020.04.028

[3] Tan W, Hao F, McIntyre RS, et al. Is Returning to Work during the COVID-19 Pandemic Stressful? A Study on Immediate Mental Health Status and Psychoneuroimmunity Prevention Measures of Chinese Workforce. Brain Behav Immun. 2020;S0889-1591(20)30603-6. doi:10.1016/j.bbi.2020.04.055

[4] Hao F, Tan W, Jiang L, et al. Do psychiatric patients experience more psychiatric symptoms during COVID-19 pandemic and lockdown? A Case-Control Study with Service and Research Implications for Immunopsychiatry. Brain Behav Immun. 2020;S0889-1591(20)30626-7. doi:10.1016/j.bbi.2020.04.069

[5] Tran BX, Phan HT, Nguyen TPT, et al. Reaching further by Village Health Collaborators: The informal health taskforce of Vietnam for COVID-19 responses. J Glob Health. 2020;10(1):010354. doi:10.7189/jogh.10.010354

2) For the following statement, the authors need to cite references:

Under this background, healthcare workers, especially the first line healthcare workers, are faced with unimaginable challenges.

Reference:

Tan BYQ, Chew NWS, Lee GKH, et al. Psychological Impact of the COVID-19 Pandemic on Health Care Workers in Singapore [published online ahead of print, 2020 Apr 6]. Ann Intern Med. 2020;M20-1083. doi:10.7326/M20-1083

Reply: 

We have revised the descriptions and included the reference in the revision as below: 

- Page 3, line 44-45: "Under this background, healthcare workers, especially the first line healthcare workers, are faced with unimaginable challenges [6]".

References: 

[6] Tan BYQ, Chew NWS, Lee GKH, et al. Psychological Impact of the COVID-19 Pandemic on Health Care Workers in Singapore. Ann Intern Med. 2020;M20-1083. doi:10.7326/M20-1083

3) Under the discussion, the authors stated "Literature revealed that SARS in 117 2003, Wenchuan earthquake in 2008, Ebola virus in 2014, and MERS, all these emergency 118 events caused different degrees of psychological trauma to the healthcare workers, including anxiety, insomnia, depression and emotional disorders [2-6]" 

Please compare their findings with COVID-19 study instead:

Chew NWS, Lee GKH, Tan BYQ, et al. A multinational, multicentre study on the psychological outcomes and associated physical symptoms amongst healthcare workers during COVID-19 outbreak [published online ahead of print, 2020 Apr 21]. Brain Behav Immun. 2020;S0889-1591(20)30523-7. doi:10.1016/j.bbi.2020.04.049

Reply: 

We have revised the descriptions and included the reference in the revision as below: 

- Page 10, line 118-123: " Literature revealed that SARS in 2003, Wenchuan earthquake in 2008, Ebola virus in 2014, and MERS, all these emergency events caused different degrees of psychological trauma to the healthcare workers, including anxiety, insomnia, depression and emotional disorders [7-10]. With more infected cases as well as death cases in the COVID-19 epidemic, healthcare workers have been facing more challenges physically and psychologically[11]".

References: 

[11] Chew NWS, Lee GKH, Tan BYQ, et al. A multinational, multicentre study on the psychological outcomes and associated physical symptoms amongst healthcare workers during COVID-19 outbreak. Brain Behav Immun. 2020;S0889-1591(20)30523-7. doi:10.1016/j.bbi.2020.04.049

---

## [Decision Letter · Decision Letter 1]

1 Dec 2020

Investigation of anxiety levels of 1637 healthcare workers during the epidemic of COVID-19

PONE-D-20-20196R1

Dear Dr. Junfen Fu,

We’re pleased to inform you that your manuscript has been judged scientifically suitable for publication and will be formally accepted for publication once it meets all outstanding technical requirements.

Kind regards,

Shawky M. Aboelhadid, PhD

Academic Editor

PLOS ONE

Additional Editor Comments (optional):

Reviewers' comments:

Reviewer's Responses to Questions

**Comments to the Author**

1. If the authors have adequately addressed your comments raised in a previous round of review and you feel that this manuscript is now acceptable for publication, you may indicate that here to bypass the “Comments to the Author” section, enter your conflict of interest statement in the “Confidential to Editor” section, and submit your "Accept" recommendation.

Reviewer #1: All comments have been addressed

Reviewer #2: All comments have been addressed

2. Is the manuscript technically sound, and do the data support the conclusions?

Reviewer #1: Yes

Reviewer #2: Yes

3. Has the statistical analysis been performed appropriately and rigorously? 

Reviewer #1: Yes

Reviewer #2: Yes

4. Have the authors made all data underlying the findings in their manuscript fully available?

Reviewer #1: Yes

Reviewer #2: Yes

5. Is the manuscript presented in an intelligible fashion and written in standard English?

Reviewer #1: Yes

Reviewer #2: Yes

6. Review Comments to the Author

Reviewer #1: The authors have addressed this reviewer's concerns. I have no further comments/edits/revisions to this manuscript.

Reviewer #2: Thanks you for addressing the comments. I have reviewed the paper and I recommend publication for this paper.

7. PLOS authors have the option to publish the peer review history of their article (what does this mean?). If published, this will include your full peer review and any attached files.

Reviewer #1: No

Reviewer #2: **Yes: **Roger Ho

---

## [Editor Report · Acceptance letter]

14 Dec 2020

PONE-D-20-20196R1 

Investigation of anxiety levels of 1637 healthcare workers during the epidemic of COVID-19 

Dear Dr. fu:

I'm pleased to inform you that your manuscript has been deemed suitable for publication in PLOS ONE. Congratulations! Your manuscript is now with our production department. 

Kind regards, 

on behalf of

Professor Shawky M. Aboelhadid 

Academic Editor

PLOS ONE